# Embedded Spatial–Temporal Convolutional Neural Network Based on Scattered Light Signals for Fire and Interferential Aerosol Classification

**Fang Xu [1], Ming Zhu [2,3], Mengxue Lin [2,3,*] , Maosen Wang [2,3] and Lei Chen [2,3]**

[1] Shenyang Fire Research Institute of M.E.M., Shenyang 110034, China; xf731110@163.com
[2] Hubei Key Laboratory of Smart Internet Technology, School of Electronic Information and Communications, Huazhong University of Science and Technology, Wuhan 430074, China; zhuming@mail.hust.edu.cn (M.Z.); m202372937@hust.edu.cn (M.W.); m202272488@hust.edu.cn (L.C.)
[3] National Engineering Research Center of Fire and Emergency Rescue, Wuhan 430074, China
* Correspondence: lin_mx@hust.edu.cn

**Abstract:** Photoelectric smoke detectors are the most cost-effective devices for very early warning fire alarms. However, due to the different light intensity response values of different kinds of fire smoke and interference from interferential aerosols, they have a high false-alarm rate, which limits their popularity in Chinese homes. To address these issues, an embedded spatial–temporal convolutional neural network (EST-CNN) model is proposed for real fire smoke identification and aerosol (fire smoke and interferential aerosols) classification. The EST-CNN consists of three modules, including information fusion, scattering feature extraction, and aerosol classification. Moreover, a two-dimensional spatial–temporal scattering (2D-TS) matrix is designed to fuse the scattered light intensities in different channels and adjacent time slices, which is the output of the information fusion module and the input for the scattering feature extraction module. The EST-CNN is trained and tested with experimental data measured on an established fire test platform using the developed dual-wavelength dual-angle photoelectric smoke detector. The optimal network parameters were selected through extensive experiments, resulting in an average classification accuracy of 98.96% for different aerosols, with only 67 kB network parameters. The experimental results demonstrate the feasibility of installing the designed EST-CNN model directly in existing commercial photoelectric smoke detectors to realize aerosol classification.

**Keywords:** embedded spatial–temporal convolutional neural network (EST-CNN); optical scattering; fire smoke; interferential aerosols; aerosol classification

## 1. Introduction

Very early fire detection and alarm systems are of great importance for disaster risk reduction. They prevent loss of life and reduce the economic and material impact of disasters. Since the release of smoke is the most obvious characteristic of very early fire [1], fire smoke monitoring is considered to be the most effective means of fire warning. Therefore, the most common commercial fire detectors available are mostly based on smoke detection, such as image-based [2–5] and photoelectric smoke detectors [6,7]. Image-based smoke detection technologies determine the presence of smoke and the occurrence of fire in a target area by analyzing and processing video image information captured with a camera [8,9]. These methods focus on a large target area, and the smoke recognition algorithms are computationally intensive, resulting in a large overhead hardware for the detection system, which is usually used in outdoor areas such as forests [10]. In contrast, photoelectric smoke detectors, which are more sensitive and responsive to smoke, are better suited for indoor fire rapid alarms [11]. At present, they are widely used in households due to the simple detection principle and their small size and the low price of the detection core

components. However, existing smoke detection techniques actually monitor all aerosols, including real fire smoke and interferential aerosols, and do not have the ability to identify real fire smoke. This leads to a high incidence of false detector alarms [12]. For image-based detectors, images sent to the monitor can be used to identify real and fake fires remotely and manually. Due to the rapid development of computer vision, deep-learning-based techniques for detecting and recognizing real fire smoke in images are well established [13], though these techniques are extremely demanding in terms of computing performance. As for photoelectric detectors, when an alarm is set off by interfering aerosols, the fire cannot be recognized remotely as false in time, leading to a waste of rescue resources.

Photoelectric smoke detection technology is based on the optical scattering theory, in which scattered light intensity is related to the size, refractive index, shape, and concentration of the particles [14–16]. However, to simplify the detection principle, it is often assumed that the intensity of the scattered light obtained by the receiver is approximately proportional to the concentration of aerosol particles entering the measurement area in practical applications [14]. Then, a fire is considered to have occurred when the smoke concentration increases to the point where the scattered light intensity reaches the alarm threshold [15]. In fact, non-fire interfering aerosol concentrations can also cause the scattered light intensity to reach the alarm threshold if it is high enough, thereby triggering a false fire alarm [17]. Obviously, recognizing and classifying aerosols of real fire smoke and interfering aerosols is an effective way to reduce false detector alarms. In addition, even for real fire combustion smoke, the intensity of scattered light at the same concentration varies by smoke type. This means that a preset threshold of fixed scattered light intensity is not satisfactory for timely and accurate alarms in all fire situations, whereas it is helpful to increase the detector's alarm accuracy by considering types of smoke and then adaptively adjusting the threshold of scattered light intensity that triggers the alarm for each type of smoke. In conclusion, to reduce the rate of false and missed alarms for fires, photoelectric smoke detectors need to have the ability to classify aerosols.

Chaudhry et al. [18] developed a system for obtaining the scattered and transmitted light intensity of fire smoke with five wavelengths in the deep ultraviolet (UV) to near-infrared range to identify burning material based on a Random Forest algorithm. This method is impractical for commercial photoelectric detectors due to the large size and complexity of the light intensity measurement device and the requirement of scattering information in the deep UV for classification. Qu et al. [19] classified four classes of European standard fires and typical interfering aerosols using a combination of multiple parameters, such as temperature, smoke, and CO concentration. Similarly, Yu et al. [20] proposed multi-detector, real-time fire alarm technology to classify oil fumes and multiple types of real fire smoke. These methods require multiple detectors to work simultaneously, which increases the cost of detection. Liu et al. [21] proposed the use of detection information from multiple smoke detectors that are already spatially interconnected to determine whether there is a real fire based on Bayesian estimation. This method also relies on multiple smoke detectors being installed in a connected space and cannot classify different kinds of fire smoke. Zheng et al. [22] used the parameter of aerosol asymmetry ratio at two wavelengths and two angles to classify black smoke, white smoke, and interference aerosols. However, this method can only distinguish dust, which has significantly different physical characterization parameters from fire smoke, and cannot identify oil fumes. In summary, aerosol (fire smoke and interferential aerosols) recognition and classification methods based on multiple channels and multiple detectors using machine learning and deep learning algorithms have attracted extensive attention. However, practical applications require methods that can be directly applied to existing single photoelectric detectors without additional overhead hardware. In detail, a practical approach needs to satisfy the following two requirements: First, the computational complexity of the classification algorithm needs to be so low that it can be used with common commercial detectors. Second, the data samples required for classification cannot exceed the actual number of channels in the detector.

To satisfy these demands, an embedded spatial–temporal convolutional neural network (EST-CNN) model for fire smoke recognition and classification is proposed. The EST-CNN model consists of three modules: information fusion of spatial–temporal scattered light intensity, scattering feature extraction, and aerosol classification. In the information fusion module, the two dimensional spatial–temporal scattering (2D-TS) matrix of the aerosol is obtained based on the scattered light intensities in different channels and adjacent time slices. The 2D-TS matrix is the input of the feature extraction module consisting of multi-layer convolutional neural networks (CNN), and the output features are based on a fully connected network (FCN) for aerosol classification. Moreover, a dual-wavelength dual-angle smoke detector is developed to acquire the aerosol datasets on combustion experimental platform for training and testing the EST-CNN model. The contributions of this study are as follows.

(1) An EST-CNN model that can be directly used in existing commercial photoelectric smoke detectors is established for interferential aerosol recognition and real fire classification.

(2) A 2D-TS matrix is created to describe the smoke scattering distribution information in spatial and temporal to obtain sufficient characterization parameters during aerosol generation.

(3) Methods for constructing and pre-processing the scattered light intensity datasets of real fire smoke and interferential aerosol are provided.

(4) The detector and experimental platform are designed to measure the scattered light intensity information of standard fire smoke and interference oil fumes.

The remaining parts of the paper are organized as follows: Section 2 introduces the mechanism of aerosol detection, the datasets used for classification, and the EST-CNN model. Section 3 presents the experimental platform for acquiring the datasets and discusses the performance of the EST-CNN model. Section 4 is the conclusion of this study.

## 2. Materials and Methods

### 2.1. Aerosol Optical Classification Mechanism

Photoelectric smoke detection technology is based on the Mie scattering theory [23], in which the scattered light intensity $I_S$ is related to the incident light intensity $I_0$, dimensionless particle size $\alpha$ (the ratio of the particle size $x$ to the incident light wavelength $\lambda$), and the scattering angle $\theta$, as shown in Equation (1).

$$I_S = \frac{\lambda^2}{8\pi^2 r^2} |S(m, \alpha, \theta)|^2 I_0, \tag{1}$$

where $r$ is the distance between the receiver to the particle, $m$ is the refractive index, and $S(m, \alpha, \theta)$ represents the amplitude function of the scattered light. It can be seen that in Equation (1), $x$ and $m$ are determined by the original characteristics of the particles, while $\lambda$, $\theta$, and $r$ are determined by the design of the detector. Generally, $r$ is the coefficient of the scattered light intensity, which is taken as a fixed constant.

However, the actual scattered light intensity $I_{SR}$ of real-life aerosol is controlled by multiple variables simultaneously, such as particle size distribution (PSD). The production of fire smoke (or oil fumes) is a process of continuous aerosol generation and aggregation. The size of the freshly generated particles is very small and gradually increases with aggregation. Thus, the particle size of aerosol obeys a distribution $f(x)$ rather than the monodisperse system with single value. Then, $I_{SR}$ is written as Equation (2).

$$I_{SR} = N \sum_{x_{\min}}^{x_{\max}} I_S(\lambda, \theta, x, m) f(x), \tag{2}$$

where $N$ is the total number concentration of real-life aerosol, $(x_{\min}, x_{\max})$ is the particle size range, and $I_S(\lambda, \theta, x, m)$ denotes the scattered light intensity of a single particle with size $x$. According to Equations (1) and (2), the inherent property parameters $(m, f(x))$ of different classes of aerosol particles are variable, leading to differences in scattered light

intensity $I_{SR}$. This means that the aerosol class information is involved in the scattered light intensity; however, it cannot be directly separated. Thus, the multi-channel scattered light intensity information feature analysis method is utilized for aerosol classification, which is based on the specificity in the scattering features of each class of aerosol under different conditions. To apply this method, the smoke detector receivers are required to obtain the scattered light intensity at different wavelengths of incident light and scattering angles.

### 2.2. Dataset for Classification

The smoke from the real fires used in this study complies with European standard fires, such as beech wood smoldering fire (TF2), cotton smoldering fire (TF3), polyurethane open flame (TF4), and n-heptane open flame (TF5) [19,24–28]. Moreover, a survey by the National Fire Protection Association (NFPA) reported that the most likely cause of false alarms by detectors is oil fumes because their particle sizes as well as their refractive indices being very close to that of real fire smoke [29]. As a result, oil fumes are used as one class of typical interferential aerosols. In addition, water mist and dust, which are the most frequent causes of false alarms in day-to-day life, also act as interferential aerosols. For aerosols, scattered light intensity under different scattering channels constitutes spatial feature vector data. Considering that the proposed classification method is expected to be directly applicable to existing photodetectors, data from four detection channels with dual wavelengths and dual angles are used. Moreover, as mentioned in the previous subsection, aerosols are continuously generated and aggregated during measurement; thus, the scattered light intensity at different times constitutes the temporal feature vector data. To ensure that the scattering feature matrix is a square matrix, the temporal feature vector has the same length as the spatial feature vector, i.e., it consists of the scattered light intensity data at the current time point and the three previous time points. The classes of the aerosols (real fire smoke and interferential aerosols) and each aerosol dataset used for classification are shown in Tables 1 and 2. The labels in Table 1 show the seven aerosol classes (four standard real fire smoke classes and three interferential aerosols) in the classification task of the network. As shown in Table 2, the elements of the feature dataset matrix $D^i_{mn}$ represent the scattered light intensity of the detector under the $n_{\text{th}}$ optical channel at the $m_{\text{th}}$ time point for the $i_{\text{th}}$ aerosol.

**Table 1.** The class of real fire smoke and interferential aerosol.

| Aerosol | Beech Smoke (TF2) | | Cotton Smoke (TF3) | | Polyurethane Smoke (TF4) | | N-Heptane Smoke (TF5) | | Oil Fume (Interferential Aerosol) | | Dust (Interferential Aerosol) | | Water Mist (Interferential Aerosol) | |
|---|---|---|---|---|---|---|---|---|---|---|---|---|---|---|
| Label | 0 | | 1 | | 2 | | 3 | | 4 | | 5 | | 6 | |
| Feature dataset | $\begin{bmatrix} D^1_{11} & \cdots & D^1_{14} \\ \vdots & \ddots & \vdots \\ D^1_{41} & \cdots & D^1_{44} \end{bmatrix}$ | | $\begin{bmatrix} D^2_{11} & \cdots & D^2_{14} \\ \vdots & \ddots & \vdots \\ D^2_{41} & \cdots & D^2_{44} \end{bmatrix}$ | | $\begin{bmatrix} D^3_{11} & \cdots & D^3_{14} \\ \vdots & \ddots & \vdots \\ D^3_{41} & \cdots & D^3_{44} \end{bmatrix}$ | | $\begin{bmatrix} D^4_{11} & \cdots & D^4_{14} \\ \vdots & \ddots & \vdots \\ D^4_{41} & \cdots & D^4_{44} \end{bmatrix}$ | | $\begin{bmatrix} D^5_{11} & \cdots & D^5_{14} \\ \vdots & \ddots & \vdots \\ D^5_{41} & \cdots & D^5_{44} \end{bmatrix}$ | | $\begin{bmatrix} D^6_{11} & \cdots & D^6_{14} \\ \vdots & \ddots & \vdots \\ D^6_{41} & \cdots & D^6_{44} \end{bmatrix}$ | | $\begin{bmatrix} D^7_{11} & \cdots & D^7_{14} \\ \vdots & \ddots & \vdots \\ D^7_{41} & \cdots & D^7_{44} \end{bmatrix}$ | |

**Table 2.** The dataset of aerosol beech smoke used for classification.

| Beech Smoke | Optical Channel 1 | Optical Channel 2 | Optical Channel 3 | Optical Channel 4 |
|---|---|---|---|---|
| **Time 1** | $D^1_{11}$ | $D^1_{12}$ | $D^1_{31}$ | $D^1_{41}$ |
| **Time 2** | $D^1_{21}$ | $D^1_{22}$ | $D^1_{32}$ | $D^1_{42}$ |
| **Time 3** | $D^1_{31}$ | $D^1_{32}$ | $D^1_{33}$ | $D^1_{43}$ |
| **Time 4** | $D^1_{41}$ | $D^1_{42}$ | $D^1_{34}$ | $D^1_{44}$ |

### 2.3. Embedded Spatial–Temporal Convolution Neural Network

The EST-CNN model consists of three modules: data preprocessing, feature analysis, and classification. The data preprocessing and feature analysis modules use CNN and the classification module uses FCN. The overall flow diagram and network logic schematic of the EST-CNN model are shown in Figures 1 and 2, respectively. Limited by the processor

performance of the photoelectric smoke detector, the overall number of parameters of the model is required to be as limited as possible, i.e., the number of layers and nodes of the network are required to be tailored as much as possible. Therefore, in data preprocessing, both the spatial and temporal data of the scattered light intensity are extracted with one layer of CNN to extract the corresponding feature vectors, respectively. The scattering feature extraction network is composed of three CNNs with convolutional kernels of the following sizes: $3 \times 3$, $3 \times 3$, and $2 \times 2$. Aerosol classification network is a single fully connected layer with a SoftMax function.

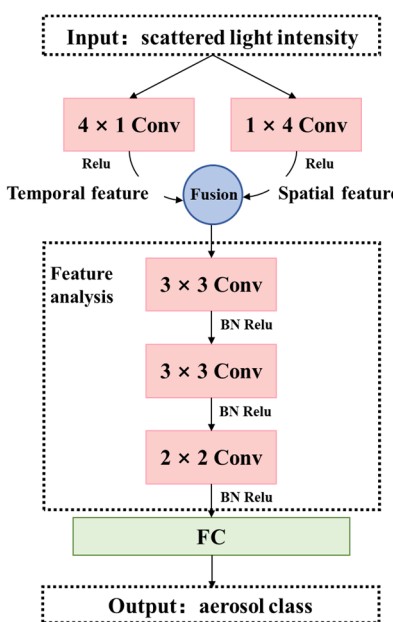

**Figure 1.** The overall flow diagram of EST-CNN.

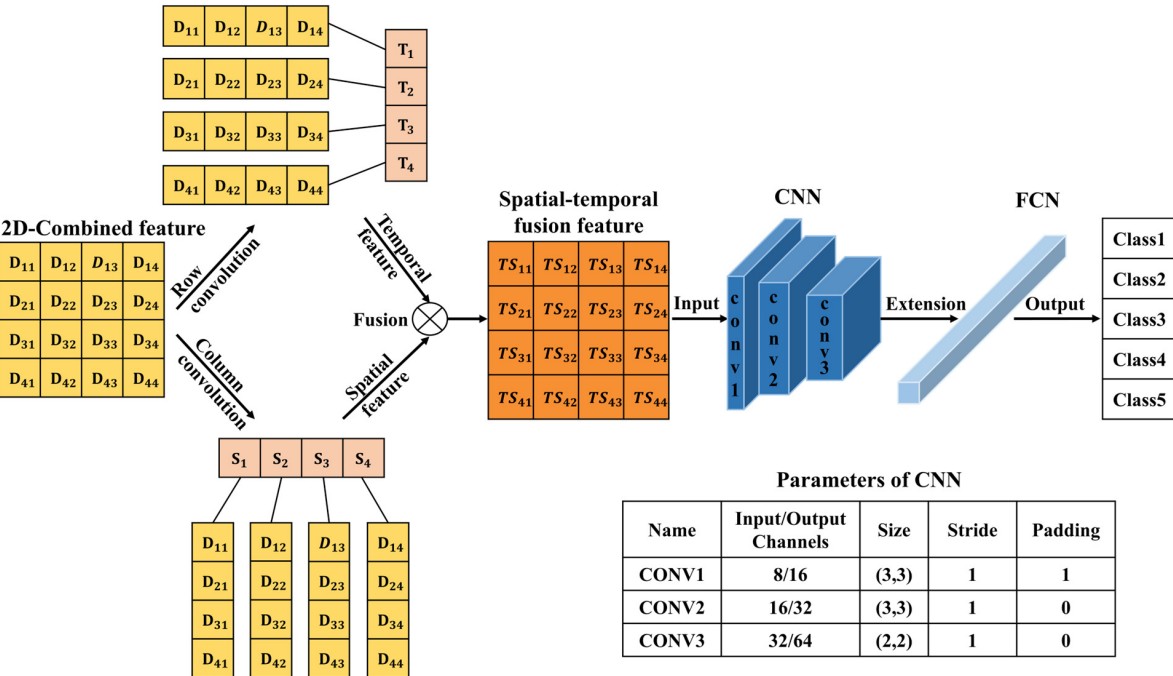

**Figure 2.** The logic schematic of EST-CNN $\begin{bmatrix} D_{11} & D_{12} & D_{13} & D_{14} \\ D_{21} & D_{22} & D_{23} & D_{24} \\ D_{31} & D_{32} & D_{33} & D_{34} \\ D_{41} & D_{42} & D_{43} & D_{44} \end{bmatrix}$.

**Parameters of CNN**

| Name | Input/Output Channels | Size | Stride | Padding |
|------|-----------------------|------|--------|---------|
| CONV1 | 8/16 | (3,3) | 1 | 1 |
| CONV2 | 16/32 | (3,3) | 1 | 0 |
| CONV3 | 32/64 | (2,2) | 1 | 0 |

As shown in Figure 2, the 2D-Data matrix recodes the original scattered light intensity, whose row and column vectors are convolved to obtain the temporal and spatial features vector of the scattered light intensity. Then, the spatial–temporal fusion scattering feature matrix (2D-TS matrix) is obtained by the vector outer product. The 2D-TS matrix is the input of the feature extraction network which contains three CNNs, each consisting of a convolutional layer, a normalization layer, and a rule activation layer. The last level output is extended to a 1D vector as the input of FCN. The network model is trained and tested with the actual measurements of scattered light intensity from photoelectric smoke detectors. The *PR* curve is used to evaluate the training performance of the model, and the larger area of the curve and the area enclosed by the coordinate axis indicates the better training performance. The confusion matrix heat map is used to demonstrate the classification accuracy of the trained model for smoke (oil fumes). On the *PR* curve, *P* and *R* are the accuracy and recall of the classification prediction results, respectively, which are defined as shown in Equations (3) and (4).

$$P = \frac{TP}{TP + FP}, \tag{3}$$

$$R = \frac{TP}{TP + FN}, \tag{4}$$

where $TP$, $FP$, and $FN$ represent true positive, false positive, and false negative, respectively. In fire alarms, they represent true fire, false alarm, and missed alarm, respectively. A larger area enclosed using the *PR* curve indicates a better balance between false alarms and missed alarms. The $F_1$ score is employed to quantify the general performance of the *PR* curve, as shown in Equation (5).

$$F_1 = \frac{2 \times P \times R}{P + R}. \tag{5}$$

Additionally, *FLOPs* is employed to evaluate the time complexity of the network. The larger the *FLOPs*, the slower the model training and inference. For CNN, the computational complexity of each convolutional layer is shown in Equation (6).

$$FLOPs\_CNN = \left[ \left( C_i \times k^2 \right) + \left( C_i \times k^2 - 1 \right) + 1 \right] \times C_o \times W \times H, \tag{6}$$

where $C_i$ and $C_o$ are the number of the input and output channels, $k$ is the convolutional kernel size, and $W$ and $H$ are the size of the feature maps. And for FCN, the computational complexity of each convolutional layer is shown in Equation (7).

$$FLOPs\_FCN = [I + (I - 1) + 1] \times O. \tag{7}$$

where $I$ and $O$ are the input and output neurons.

## 3. Results and Discussion

### 3.1. Experimental Platform and Datasets

The experimental platform and the photoelectric detector are shown in Figure 3. Figure 3a shows a complete view of the experimental area, including the experimental platform, the hood, and the photoelectric smoke detector placed on the ceiling. Figure 3b shows the physical appearance of the photoelectric smoke detector and the internal structure of the measurement chamber. Figure 3c shows a scheme of the measurement principle of a photoelectric smoke detector. The detector consists of a dual-wavelength emitter LED that can emit blue light with wavelength $\lambda_1$ and infrared light with wavelength $\lambda_2$ and three photoreceivers with the scattering angle of $\theta_1, \theta_2, \theta_3$. LED emits two wavelengths of light in sequence. The photoreceiver at the scattering angle of $\theta_1$ receives forward blue and infrared light, the photoreceiver at the scattering angle of $\theta_2$ receives infrared light, and the photoreceiver at the scattering angle of $\theta_3$ receives blue light. Thus, scattered light intensity data can be obtained for four channels, namely $(\lambda_1, \theta_1)$, $(\lambda_1, \theta_2)$, $(\lambda_2, \theta_1)$, and $(\lambda_2, \theta_3)$, for

each measurement. These channels are selected based on simulation calculations by substituting the typical characteristic parameters of seven classes (four real fire smoke types and three interferential aerosols) of aerosols based on the Mie scattering theory.

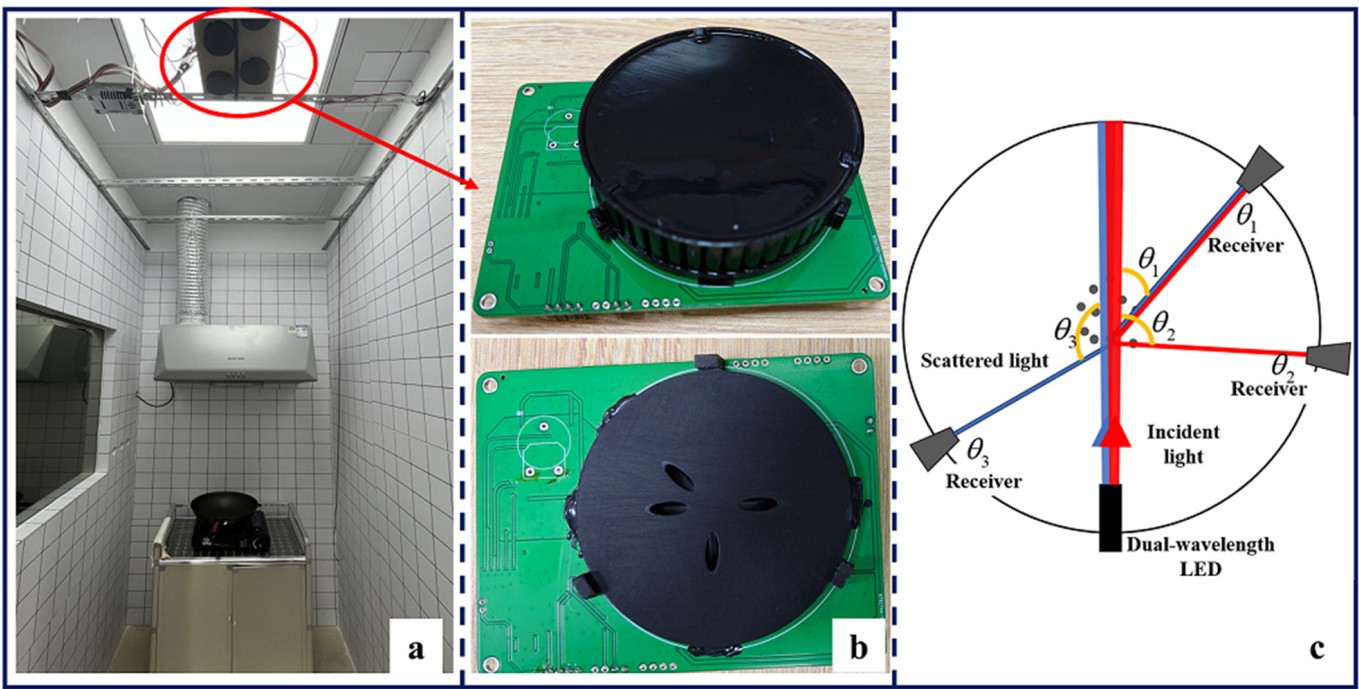

**Figure 3.** Experimental platform and the photoelectric detector. (**a**) appearance of the experimental platform, (**b**) real smoke detector, (**c**) the schematic diagram of the detector measuring fire smoke.

Real fire smoke and interferential aerosols were generated as shown in Figure 4. Beech wood smoke was obtained by heating in a resistance-heated furnace. Cotton smoke was obtained under smoldering conditions. Polyurethane and n-heptane smoke were generated by open flame combustion. Oil fumes were generated by frying minced pork. Water mist was generated by a humidifier. Dust was produced by mixing Alexander's Standard Ash in an acrylic smoke box. Smoke and interferential aerosols naturally diffused into the photoelectric smoke detector during the measurement. After each measurement, the hood was turned on to remove the aerosol generated from the experiment. The scattered light intensity value measured by the photoelectric smoke detector when there was no smoke was used as a background. The values of scattered light intensities of smoke and interferential aerosols were obtained by subtracting the background value from the scattered light intensity measurement. Experiments show that beech and cotton smolder slowly and generate a small amount of white smoke. Polyurethane and n-heptane combust rapidly and generate large amounts of black smoke with a high carbon content due to insufficient combustion. This means that the trend of the scattered light intensity over time is directly related to the class of smoke (and interferential aerosols). Thus, multiple four-channel samples of scattered light intensity are collected in chronological order for each experiment during smoke (and interferential aerosols) generation to acquire the temporal scattering feature.

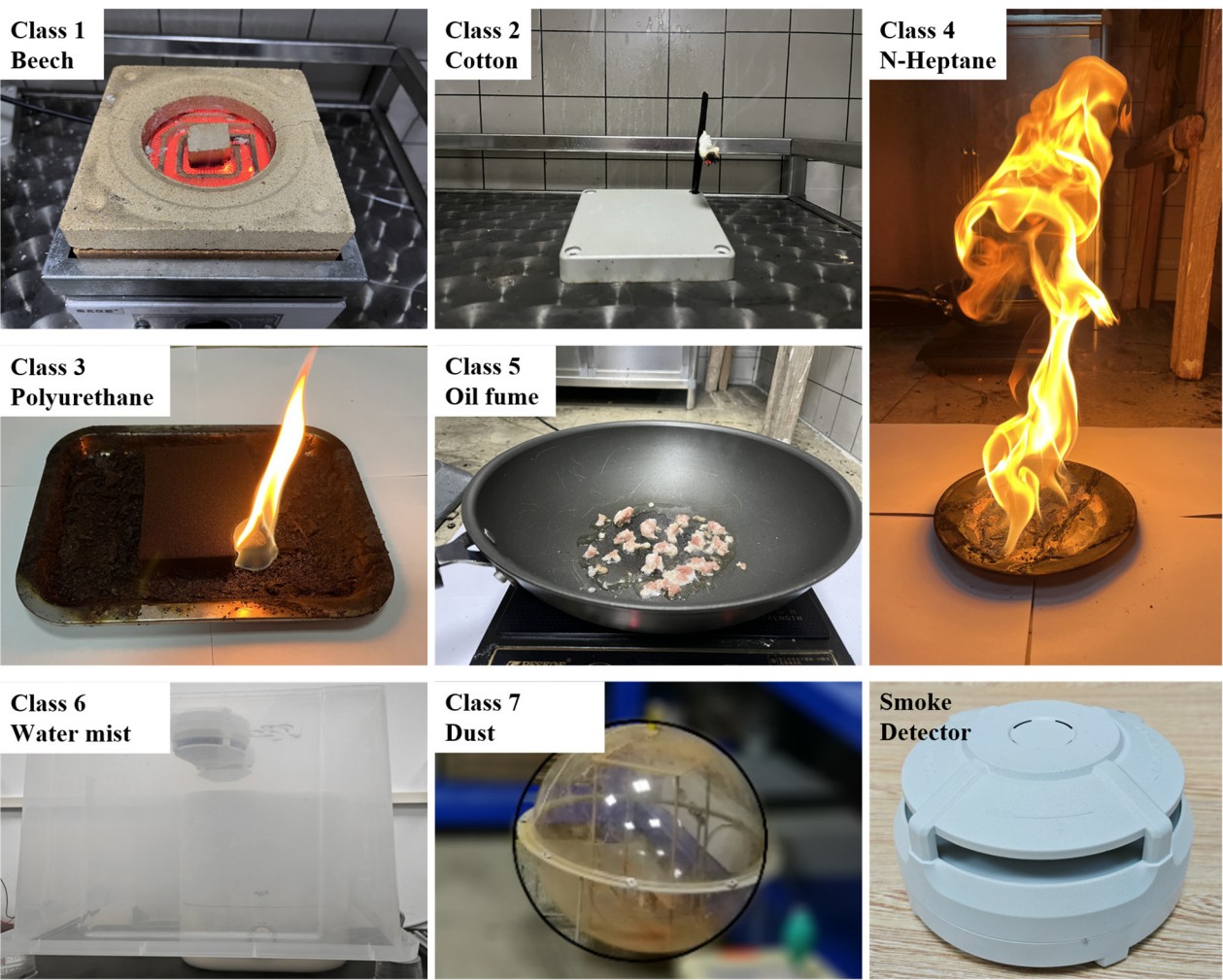

**Figure 4.** Four classes of standard real fire smoke (beech wood, cotton, polyurethane, and n-heptane smoke), three class of interferential aerosols (oil fume, water mist, and dust), and the developed smoke detector.

### 3.2. Classification Results

A total of 38,487 samples of beech, cotton, polyurethane, n-heptane smoke, oil fumes, water mist, and dust were obtained by the experimental platform for classification and were divided into training and test sets at a ratio of 7:3. The algorithms were run in PyCharm (PyCharm Community Edition 2020.1) using a PC with an Intel (R) Core (TM) and i7-10510U CPU @ 1.80 GHz 2.30 GHz. The EST-CNN model was trained and tested using multiple sets of parameters to identify the optimum ones with low numbers of network parameters and high classification accuracy. The EST-CNN model was trained for 300 epochs at each hyperparameter setting.

CNN for feature learning and extraction typically have more than three convolutional layers used to first increase and then decrease the dimensionality of the data. To realize the model running on low-computing power-embedded chips, the number of parameters of the model is required to be as small as possible. Therefore, the EST-CNN model applies only three convolutional layers in the feature extraction network, and the input data size of the first layer is 4 (number of optical channels) × 4 (sampling time points). The previous input and final output channels of the feature extraction network are connected with the 2D-TS matrix and full connected classification layer, respectively, which are set to constant values of 8 and 64. In this way, the main parameters that can be adjusted in the model are the number of input and output channels in the middle layer, stride, and padding. For the channels, the number of output channels in each layer of the network is required

to be the same as the number of input channels in the next layer. Stride represents the amount of data that the convolution kernel moves over in each slide. Padding comprises the complementary zeros around the boundaries of the input matrix. The parameters of the partial sets of the feature extraction network are shown in Table 3, and the corresponding training and testing results are shown in Figures 5–9. In these figures, labeled 0, 1, 2, 3, 4, 5, and 6, represent beech smoke, cotton smoke, polyurethane smoke, n-heptane smoke, oil fumes, dust, and water mist, respectively.

**Table 3.** The parameters of the feature extraction network.

| Set | Layer Number | Layer Type | Input Channel | Output Channel | Convolutional Kernel Size | Stride | Padding | Parameters | FLOPs | Classification Accuracy |
|---|---|---|---|---|---|---|---|---|---|---|
| **1** | 1 | Conv | 8 | 16 | $3 \times 3$ | 1 | 1 | 67 kB | 0.15 M | 98.96% |
| | 2 | Conv | 16 | 32 | $3 \times 3$ | 1 | 0 | | 1.81 M | |
| | 3 | Conv | 32 | 64 | $2 \times 2$ | 1 | 0 | | 15.75 M | |
| **2** | 1 | Conv | 8 | 32 | $3 \times 3$ | 1 | 1 | 158 kB | 0.29 M | 99.04% |
| | 2 | Conv | 32 | 64 | $3 \times 3$ | 1 | 0 | | 33.18 M | |
| | 3 | Conv | 64 | 64 | $2 \times 2$ | 1 | 0 | | 130.06 M | |
| **3** | 1 | Conv | 8 | 32 | $3 \times 3$ | 1 | 1 | 295 kB | 0.29 M | 99.09% |
| | 2 | Conv | 32 | 128 | $3 \times 3$ | 1 | 0 | | 66.36 M | |
| | 3 | Conv | 128 | 64 | $2 \times 2$ | 1 | 0 | | 1057.03 M | |
| **4** | 1 | Conv | 8 | 16 | $3 \times 3$ | 1 | 1 | 67 kB | 0.15 M | 98.89% |
| | 2 | Conv | 16 | 32 | $3 \times 3$ | 2 | 1 | | 0.67 M | |
| | 3 | Conv | 32 | 64 | $2 \times 2$ | 1 | 0 | | 15.75 M | |
| **5** | 1 | Conv | 8 | 16 | $3 \times 3$ | 1 | 1 | 67 kB | 0.15 M | 98.84% |
| | 2 | Conv | 16 | 32 | $3 \times 3$ | 2 | 1 | | 0.67 M | |
| | 3 | Conv | 32 | 64 | $2 \times 2$ | 2 | 0 | | 4.19 M | |

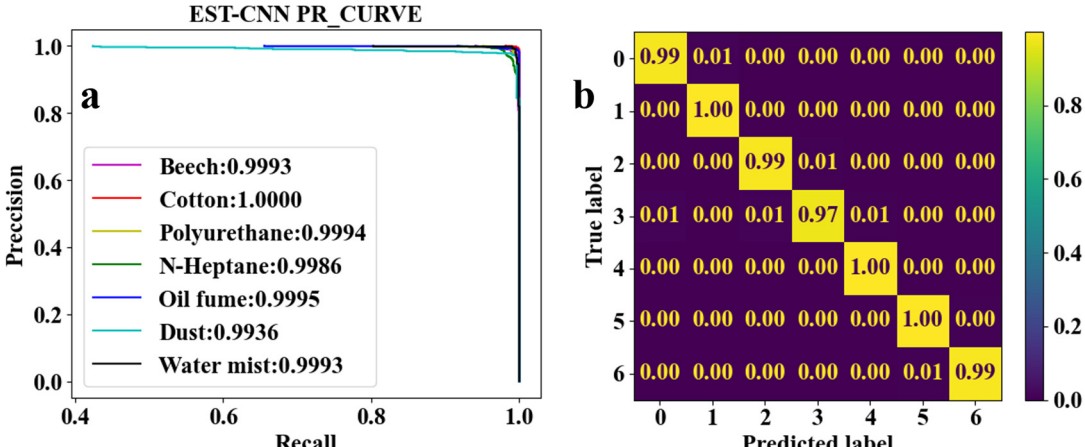

**Figure 5.** Classification results of seven classes of real fire smoke and interferential aerosols under the EST-CNN model using the **Set 1** network parameters. (**a**) Training results; (**b**) test results.

As shown in Table 3, both the first convolutional layer and the second convolutional layer of **Set 2** have more output channels than those of **Set 1**. Similarly, those of **Set 3** have more channels than those of **Set 2**. As a result, the number of parameters in **Set 2** and **Set 3** are, respectively, 2.4 and 4.5 times that of **Set 1**. However, it can be seen from Figure 5a, Figure 6a, and Figure 7a that the $F_1$ scores of **Set 1**, **Set 2**, and **Set 3** are very close to each other and all the $F_1$ scores are higher than 0.9900. This demonstrates that when trained under each set of parameters, the network model converges to a reliable classification performance. Meanwhile, as shown in Figures 5b and 6b, test results indicate that the classification accuracy of n-heptane smoke in **Set 1** is slightly lower than that in **Set 2**, and the other three classes of smoke have almost the same classification accuracy under both **Set 1** and **Set 2**. Thus, it is considered that the increased number of parameters

in **Set 2** is meaningless. Moreover, it can be seen from Figures 6b and 7b that the models under the **Set 2** and **Set 3** of hyperparameters have the same classification accuracy for real fire smoke as well as for interferential aerosols. Therefore, the number of input and output channels of each layer in **Set 1** is optimal.

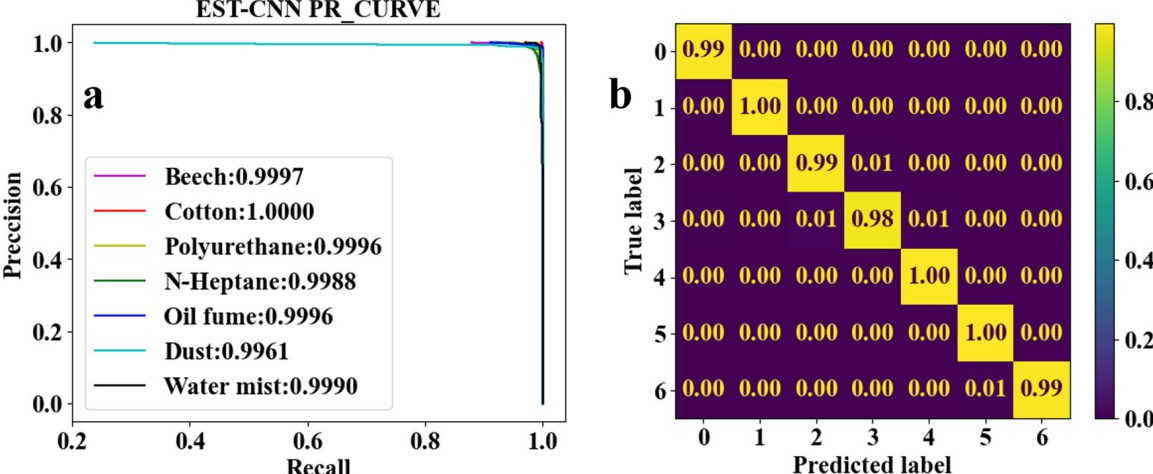

**Figure 6.** Classification results of seven classes of real fire smoke and interferential aerosols under the EST-CNN model using the **Set 2** network parameters. (**a**) Training results; (**b**) test results.

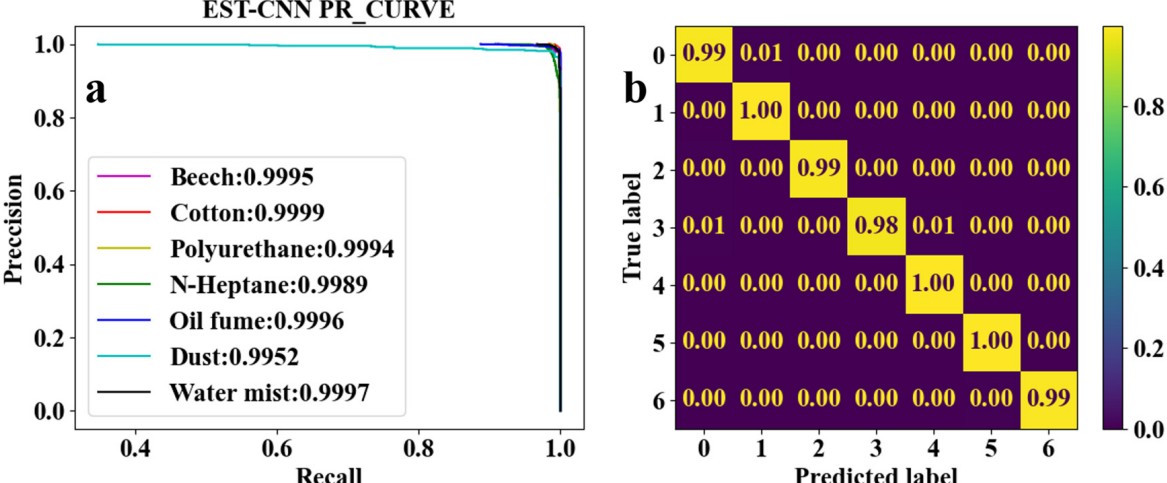

**Figure 7.** Classification results of seven classes of real fire smoke and interferential aerosols under the EST-CNN model using **the Set 3** network parameters. (**a**) Training results; (**b**) test results.

False alarms in traditional photoelectric smoke detectors occur mostly due to the misrecognition of interferential aerosols as fire smoke. Therefore, whether or not the disturbing aerosol is classified as real fire smoke is critical for evaluating the EST-CNN network. As can be seen from the confusion matrix in Figure 5b, only 1% of the n-heptane smoke (label 3) was misclassified as oil fumes (label 4) in all instances of real fire smoke (labels 0 to 3). This proves that the real fire missing alarm rate of the proposed model is only 0.25%, which is extremely low. In addition, none of the interfering aerosols (labels 4 to 6) were misclassified as real fire smoke classes. This indicates that the model classification results are able to avoid false alarms.

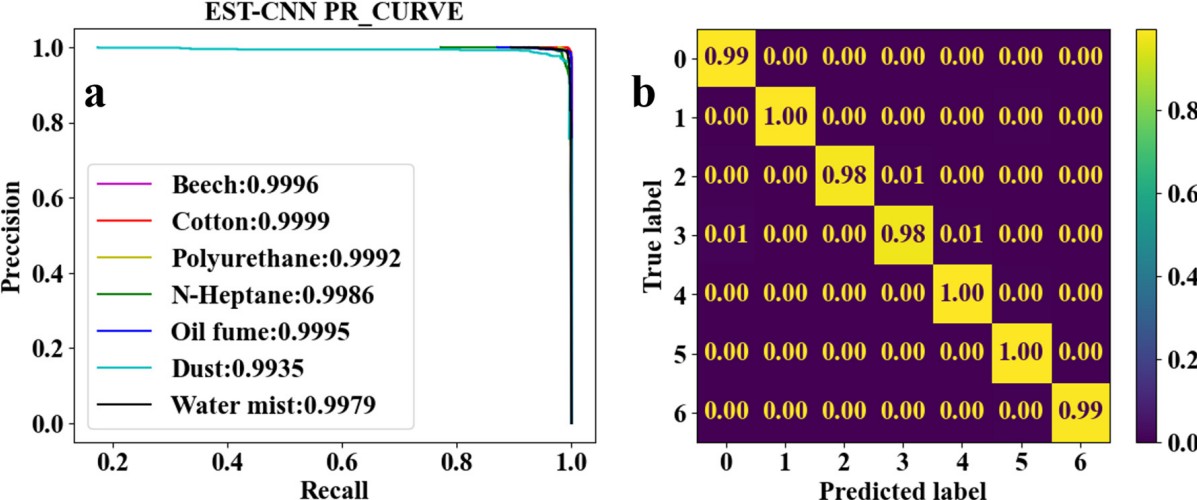

**Figure 8.** Classification results of seven classes of real fire smoke and interferential aerosols under the EST-CNN model using the **Set 4** network parameters. (**a**) Training results; (**b**) test results.

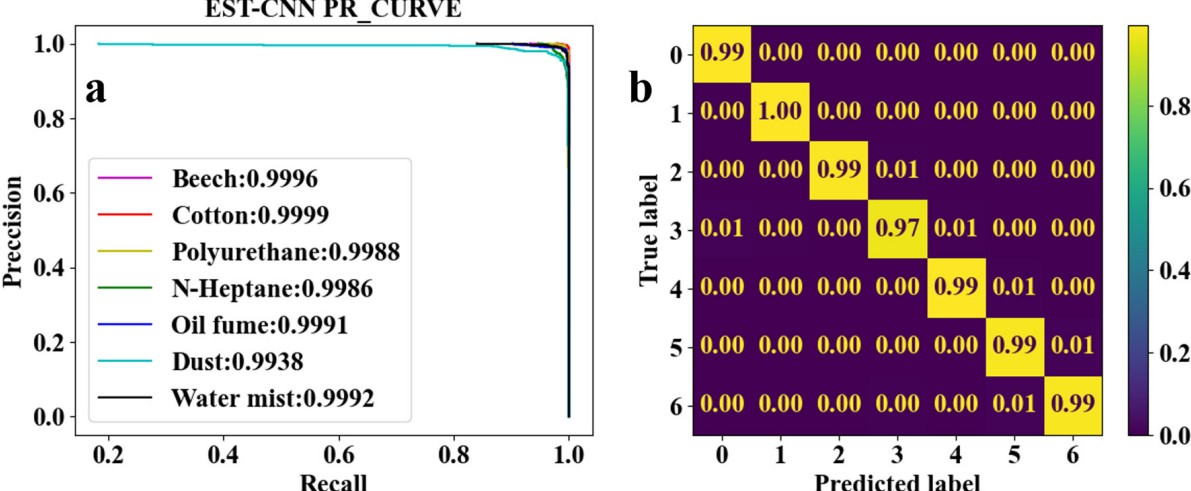

**Figure 9.** Classification results of seven classes of real fire smoke and interferential aerosols under the EST-CNN model using the **Set 5** network parameters. (**a**) Training results; (**b**) test results.

To further reduce the number of parameters, the stride and padding were adjusted, as shown in **Sets 4** and **5** in Table 3. The convolution process is essentially a multiplication of two matrices; after the convolution process, the original input matrix will have a certain degree of shrinkage; and when the stride is 1, the matrix length and width will be reduced by 2. When the length and width of the original input matrix are small (e.g., $4 \times 4$ of the 2D-TS matrix), padding the data with zeros is required before the convolution operation in order to keep the output matrix meaningful after each layer of convolution. As a result, the stride and padding usually are adjusted simultaneously, and the larger the stride the more zeros need to be padded. As shown in Table 3, **Set 4** increases the stride of only the second convolutional layer, while **Set 5** increases the stride of both the second and third convolutional layers. Nevertheless, the number of parameters in **Set 4** and **Set 5** are not reduced and remain at 66 kB. However, it can be seen in Figures 8 and 9 that the classification accuracy of oil fumes in **Set 4** and n-heptane in **Set 5** is significantly reduced compared to the accuracy of those in **Set 1,** as shown in the classification results in Figure 5.

Combining Table 3 and Figures 5–9 shows that the average classification accuracies of fire smoke and interferential aerosols under the five sets of model parameters are 98.96%, 99.04%, 99.09%, 98.89%, and 98.84%, respectively, with the parametric quantities of 67 kB,

158 kB, 295 kB, 67 kB, and 67 kB. In summary, due to the small size of the scattering feature matrix (2D-TS matrix), the parameters that can be adjusted in the convolutional layer are limited. The experimental results show that adjusting the parameters of the convolutional operation (stride and padding) has almost no influence on the number of parameters of the network model while decreasing the classification accuracy, and adjusting the number of input and output channels is an effective method to reduce the number of parameters. Therefore, the **Set 1** parameters, which exhibit the best comprehensive performance, are recognized as the actual parameters chosen for EST-CNN.

### 4. Conclusions

In this study, an embedded neural network named EST-CNN for the multi-class aerosol classification of real fire smoke and interferential aerosol is proposed. In EST-CNN, the information fusion module fuses the spatial and temporal scattered light intensity information which contains the inherent physical properties of the aerosols. To acquire the real aerosol scattered light intensity information, a detector with the same number of channels as the common photoelectric smoke detectors in the market was developed and a real fire test platform was established for experimental measurements. Then, the scattering feature extraction network with three convolutional layers was applied to obtain the most advantageous sample features in preparation for realizing aerosol classification in the last layer of FCN. The experimental results show that the classification accuracy of beech smoke, cotton smoke, polyurethane smoke, n-heptane smoke, oil fumes, dust, and water mist can reach 99%, 100%, 99%, 97%, 100%, 100%, and 100%, while the number of parameters is only at 67 kB using the network model with the selected optimal parameters.

Although the method proposed in this study has made significant contributions to the area of aerosol classification and the accurate identification of smoke from real fires, there are still many problems to be investigated in depth: for instance, how to set reasonable alarm thresholds for each class of smoke after smoke recognition and classification in order to achieve a truly very early and efficient alarm effect. In addition, the scattered light reception sensitivity of photoelectric smoke detectors from different batches made by different manufacturers is generally inconsistent, which leads to differences between scattered light intensity measurements in the same cases. In machine learning, such a situation is called distributional pair misalignment, which means that each detector requires specific model parameters, which is impractical. Therefore, our next study will adapt network models with better generalization ability.

**Author Contributions:** Conceptualization, F.X. and M.W.; methodology, M.L.; software, M.W.; validation, M.L.; formal analysis, M.W.; investigation, M.L. and L.C.; resources, M.Z.; data curation, M.W. and L.C.; writing—original draft preparation, M.L.; writing—review and editing, F.X. and M.Z.; project administration, M.Z.; funding acquisition, F.X. and M.Z. All authors have read and agreed to the published version of the manuscript.

**Funding:** This work was supported by the National Key Research and Development Program of China (grant no. 2021YFC3001600) and the National Natural Science Foundation of China (grant no. 62071189).

**Institutional Review Board Statement:** This study did not require ethical approval.

**Informed Consent Statement:** This study did not involve humans.

**Data Availability Statement:** Data can be obtained by contacting the author's e-mail address (lin_mx@hust.edu.cn).

**Conflicts of Interest:** The authors declare no conflicts of interest.

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
