# Peer review of "Embedded Spatial–Temporal Convolutional Neural Network Based on Scattered Light Signals for Fire and Interferential Aerosol Classification"

_sensors, doi:10.3390/s24030778_

Round 1
Reviewer 1 Report
Comments and Suggestions for Authors
The authors propose an embedded spatial-temporal convolutional neural network (EST-CNN) model for real fire smokes identification and aerosols (fire smokes and interferential aerosol) classification. Authors developed EST-CNN which consists of three modules including information fusion, scattering feature extraction, and aerosol classification. The authors claims that the average classification accuracy is 95.6% for different aerosols with only 66 kB network parameters.
The authors have addressed a highly sensitive problem to the societal need. I appreciate the authors for the same but below are my concerns that shall be addressed during revision:
1. Section 2 should include an overall flow diagram of the model.
2. Time complexity calculations are missing.
3. Limitations of the model to highlighted and to be included as future work.
4. Conclusion is too large. Need to be shortened.
Comments on the Quality of English LanguageMinor editing of English language required
Reviewer 2 Report
Comments and Suggestions for Authors
Lack of Novelty: The proposed approach using an embedded spatial-temporal CNN for fire smoke identification is not sufficiently novel. Similar techniques have been explored in the literature, and your paper does not significantly contribute to the existing body of knowledge.
Outdated Approach: The paper relies on the use of a dual-wavelength dual-angle photoelectric smoke detector, which is considered outdated. More recent advancements in fire detection technologies have surpassed the capabilities of dual-wavelength detectors.
Limited Applicability: The proposed EST-CNN model may face challenges in real-world implementation, particularly in environments with diverse aerosols. The applicability of the model beyond controlled experimental settings is not convincingly demonstrated.
Insufficient Comparison: The paper lacks a comprehensive comparison with existing state-of-the-art methods for fire smoke identification. Without a clear benchmarking against relevant approaches, it is challenging to assess the superiority of the proposed model.
Generalization Concerns: The limited scope of experimental data from a specific fire test platform raises concerns about the generalizability of the proposed model to various real-world scenarios.
Network Parameter Optimization: While you claim an average classification accuracy of 95.6%, the process of selecting optimal network parameters is not well-documented. A more detailed explanation and justification for parameter choices are essential for the paper's credibility.
Presentation Clarity: The paper lacks clarity in presenting key concepts, making it challenging for readers to follow the proposed model's architecture and functionality.
Inadequate Discussion on False Alarms: The paper briefly mentions the high false alarm rate of traditional photoelectric smoke detectors but does not thoroughly discuss how the proposed model addresses this issue.
Limited Practical Implications: The paper falls short in discussing the practicality and feasibility of integrating the EST-CNN model into existing commercial smoke detectors. Practical challenges and considerations are essential for the paper's relevance.
Quality of Writing: The overall quality of writing, including grammar and syntax, needs improvement to enhance the paper's readability.
Comments on the Quality of English LanguageAs part of the review process, we would like to provide feedback on the quality of the English language used in your manuscript.
Clarity of Expression: Overall, the language in the manuscript is clear and well-structured, facilitating the understanding of your proposed methodology.
Sentence Structure: The majority of sentences are well-constructed, contributing to the overall coherence of the manuscript. However, attention to sentence complexity in some sections could enhance readability.
Technical Terminology: The use of technical terms and jargon is appropriate, demonstrating a strong command of the subject matter.
Grammar and Syntax: While the manuscript is generally grammatically sound, there are instances where sentence structures could be refined for smoother flow and improved comprehension.
Verbosity: Some sections appear to be overly verbose. Consider concise phrasing to enhance the manuscript's overall clarity and reader engagement.
Consistency: Maintain consistency in the use of terminology and expressions throughout the manuscript to avoid potential confusion.
Academic Tone: The tone of the manuscript is appropriately formal and academic, aligning with the standards of scholarly writing.
Citation Format: Ensure consistent adherence to citation format guidelines, as deviations were noted in some instances.
Reviewer 3 Report
Comments and Suggestions for Authors
This article proposed an embedded spatial-temporal convolutional neural network (EST-CNN) model for fire smoke recognition and classification. The EST-CNN model consists of three modules: information fusion of spatial-temporal scattered light intensity, scattering feature extraction and aerosol classification. Obtaining the two dimensional spatial-temporal scattering (2D-TS) matrix of the aerosol by fusing the scattered light intensities from different channels and adjacent time slices. Next, the 2D-TS matrix is fed into the multi-layer convolutional neural networks (CNN) for feature extraction, and the output features are based on a fully connected network (FCN) for aerosol classification. The method appears reasonable, but some issues need to be clarified and modified. The main issues are as follows:
Major concerns:
1. Lack of explanation of the meaning of the channels in the 2D-TS matrix. As an example, what information is contained in each of the four channels shown in Table 2. Please add relevant explanations.
2. Only four classes of standard real fire smoke and one class of interferential aerosol oil fume are categorized in the article, while in reality there are far more than five categories, please increase the number of categories of aerosols in the dataset.
3. The article does not mention how many epochs the model has been trained, this parameter is important for the training of the network, so please add the relevant parameters.
4. Only one indicator was used in the experimental results section, please add evaluation indicators to demonstrate the superiority of the model.
5. In Section 3, The proposed method should be compared with the state-of-the-art methods to demonstrate its effectiveness. For example, the methods are published in 2022 and 2023. Please supplement the experiments.
Comments on the Quality of English LanguageModerate editing of English language required.
Reviewer 4 Report
Comments and Suggestions for Authors
1. The overall presentation is lacking. Strive to establish better connections between sections for a more cohesive flow. Additionally, consider utilizing grammar correction tools to enhance the writing quality.
2. Include a subsection discussing the feasibility of implementing the proposed system in real-world scenarios. Address potential challenges in terms of cost, infrastructure, and user acceptance. Discuss any pilot studies or user feedback that can validate the practicality of the developed application.
3. The paper lacks novelty overall. To substantiate the originality of your work, incorporate concrete evidence or proof points demonstrating the uniqueness of your contributions.
4. Comparitive analysis is missing.
5. What are the major contributing factors that make your model diffenent than others?
6. What are the limitations of this model when it is imple to the real time environment?
7. Consider incorporating visual aids, such as flowcharts or diagrams, to visually represent the system architecture and workflow. This enhances the paper's clarity and assists readers in comprehending the technical intricacies of the proposed solution.
8. Emphasize the user-centered design principles employed in developing the application. Discuss how the system caters to the specific needs and preferences of visually impaired users, ensuring a seamless and accessible user experience.
9. Why the authors choose EST-CNN for the classification of the fire? As there are many other CNN based models already available that can perform better.
10. On what bases the clsesses of real smoke are chosen and why only 5 classes?
Comments on the Quality of English Language
Consider utilizing grammar correction tools to enhance the writing quality.
Round 2
Reviewer 2 Report
Comments and Suggestions for Authors
Accept in present form
Reviewer 3 Report
Comments and Suggestions for Authors
Please publish this paper.
Comments on the Quality of English LanguageNone.